# Perioperative Supplemental Oxygen and Postoperative Copeptin Concentrations in Cardiac-Risk Patients Undergoing Major Abdominal Surgery—A Secondary Analysis of a Randomized Clinical Trial

**DOI:** 10.3390/jcm11082085

**Published:** 2022-04-07

**Authors:** Alexander Taschner, Barbara Kabon, Alexandra Graf, Nikolas Adamowitsch, Markus Falkner von Sonnenburg, Melanie Fraunschiel, Katharina Horvath, Edith Fleischmann, Christian Reiterer

**Affiliations:** 1Department of Anaesthesia, General Intensive Care Medicine and Pain Medicine, Medical University of Vienna, Spitalgasse 23, 1090 Vienna, Austria; alexander.taschner@meduniwien.ac.at (A.T.); barbara.kabon@meduniwien.ac.at (B.K.); nikolas.adamowitsch@meduniwien.ac.at (N.A.); markus.falknervonsonnenburg@meduniwien.ac.at (M.F.v.S.); katharina.horvath@meduniwien.ac.at (K.H.); edith.fleischmann@meduniwien.ac.at (E.F.); 2Outcome Research Consortium, Cleveland, OH 44195, USA; 3Centre for Medical Statistics, Informatics and Intelligent Systems, Medical University of Vienna, Spitalgasse 23, 1090 Vienna, Austria; alexandra.graf@meduniwien.ac.at; 4IT Systems and Communications, Medical University of Vienna, Spitalgasse 23, 1090 Vienna, Austria; melanie.fraunschiel@meduniwien.ac.at

**Keywords:** supplemental oxygen, perioperative stress, Copeptin, MINS, major abdominal surgery, cardiovascular risk

## Abstract

Noncardiac surgery is associated with hemodynamic perturbations, fluid shifts and hypoxic events, causing stress responses. Copeptin is used to assess endogenous stress and predict myocardial injury. Myocardial injury is common after noncardiac surgery, and is often caused by myocardial oxygen demand-and-supply mismatch. In this secondary analysis, we included 173 patients at risk for cardiovascular complications undergoing moderate- to high-risk major abdominal surgery. Patients were randomly assigned to receive 80% or 30% oxygen throughout surgery and the first two postoperative hours. We evaluated the effect of supplemental oxygen on postoperative Copeptin concentrations. Copeptin concentrations were measured preoperatively, within two hours after surgery, on the first and third postoperative days. In total, 85 patients received 0.8 FiO_2_, and 88 patients received 0.3 FiO_2_. There was no significant difference in postoperative Copeptin concentrations between both study groups (*p* = 0.446). Copeptin increased significantly within two hours after surgery, compared with baseline in the overall study population (estimated effect: −241.7 pmol·L^−1^; 95% CI −264.4, −219.1; *p* < 0.001). Supplemental oxygen did not significantly attenuate postoperative Copeptin release. Copeptin concentrations showed a more immediate postoperative increase compared with previously established biomarkers. Nevertheless, Copeptin concentrations did not surpass Troponin T in early determination of patients at risk for developing myocardial injury after noncardiac surgery.

## 1. Introduction

During recent years, the prevalence of cardiovascular risk factors and comorbidities in patients undergoing noncardiac surgery has significantly increased [1]. As a consequence, the incidence of postoperative major cardiovascular complications has risen to approximately 8% among this patient population [2,3].

Surgery and anesthesia are associated with trauma, hemodynamic perturbations, fluid shifts, stress and hypoxic events [4,5]. These are trigger factors for endogenous stress, reflected by increased catecholamine and cortisol release, and myocardial injury [6]. Elevated stress levels are associated with increased sympathetic nerve activity leading to tachycardia and hypertension [7]. This might lead to an imbalance in myocardial oxygen supply and demand, and finally result in myocardial injury [7,8,9]. It is very well known that perioperative hypoxic events caused by hypovolemia, hypotension, tachycardia and hypoxemia significantly increase the risk for myocardial injury after noncardiac surgery (MINS) [9,10]. A previous study has shown that preoperative Copeptin concentrations might be able to predict myocardial injury in the immediate perioperative period [11].

Copeptin is a relatively novel biomarker and reflects plasma concentrations of arginine-vasopressin (AVP) [12]. AVP is an antidiuretic hormone released from the hypothalamus in response to changes in plasma osmolality and blood pressure, and its main function is homeostasis of fluid balance, vascular tonus and regulation of the endocrine stress response [13]. In detail, an increase in blood osmolality and hypovolemia leads to increased plasma AVP and Copeptin concentrations [14]. In contrast to AVP, plasma concentrations of Copeptin are very stable and simple to measure, and are therefore used to indirectly assess plasma AVP concentration [13]. Copeptin concentrations significantly correlate with physiologic as well as pathophysiologic endogenous stress, such as that caused by surgery [13,15]. In detail, Copeptin concentrations are significantly increased in patients who have suffered from myocardial infarction, heart failure, shock, stroke and traumatic brain injury [16,17,18,19]. Elevated concentrations are explained by exacerbated endogenous stress associated with cardiovascular complications [11]. Furthermore, preoperative elevated Copeptin values are strong predictors for MINS [11]. Copeptin concentrations accurately reflect myocardial strain and injury as well as endogenous stress, and could therefore be of high value in properly reflecting perioperative stress.

In our main trial, we investigated the effect of 80% versus 30% perioperative oxygen administration on postoperative maximum NT-proBNP concentrations and MINS [20]. We observed no significant difference between both study groups [21]. Because there is limited data in regard to perioperative Copeptin concentrations, specifically on the subject of supplemental oxygen, we evaluated in this secondary analysis if supplemental oxygen influences perioperative Copeptin concentrations. Thus, we evaluated the hypothesis if perioperative administration of 80% oxygen leads to a significant decrease in postoperative Copeptin concentrations as compared to perioperative administration of 30% oxygen in patients at risk for cardiovascular complications undergoing moderate- to high-risk major abdominal surgery. Furthermore, we evaluated the effect of surgery per se as well as MINS on perioperative Copeptin concentrations in the overall study population. In a post-hoc analysis, we evaluated the predictive values of Copeptin concentrations in the perioperative time course for the development of MINS.

## 2. Materials and Methods

### 2.1. Study Design

This is a pre-planned secondary analysis of a prospective, randomised, double-blinded, single-centre clinical trial conducted at the Medical University of Vienna, which primarily investigated the effect of 80% versus 30% inspired oxygen concentration on postoperative maximum NT-proBNP concentrations [21]. This study was approved by the University’s Ethics Committee (Ethikkomission Medizinische Universität Wien; Borschkegasse 8b/6, 1090, Vienna, Austria; EK-Number 1744/2017; Chairperson Prof. Martin Brunner) on 13 November 2017. Written informed consent was obtained from all patients participating in the study. The trial was registered prior to patient enrolment at ClinicalTrials.gov (NCT03366857, Principal Investigator: Edith Fleischmann, Date of registration: 2 December 2017) and the European Trial Database (EudraCT 2017-003714-68), and was conducted according to the Declaration of Helsinki and Good Clinical Practice. This manuscript adheres to the applicable CONSORT guidelines. The study protocol was published previously [20]. The additional measurement of Copeptin concentrations for this secondary analysis was amended on 19 July 2018 after 87 patients had already been included. Patients of at least 45 years of age and undergoing major abdominal surgery for ≥2 h were eligible for the trial. Detailed inclusion and exclusion criteria were published previously [20].

### 2.2. Randomisation

For patient randomisation of the main study, a web-based randomisation programme (Randomizer, Medical University of Graz, Graz, Austria, https://www.meduniwien.ac.at/randomizer/web) (last accessed on 5 November 2019) was used. Randomisation sequence was generated by the study statistician using permutated blocks with a size of six numbers. We did not use stratification of randomisation. 

Patients were randomised to receive either 80% or 30% inspired oxygen concentration throughout surgery, and for the first two postoperative hours. We randomised patients shortly before induction of anesthesia. The trial was conducted in accordance with the original protocol [20]. Protocol for induction and maintenance of anesthesia was published previously [20]. Intraoperative fluid management in all patients was performed in an esophageal-Doppler-guided manner according to a previously published algorithm [22,23]. As per study protocol, all patients received a 2 mL·kg^−1^·BW^−1^ baseline infusion of balanced crystalloids. A bolus of 250 mL balanced crystalloids was administered when stroke volume decreased by ≥20% from baseline. In case of acute bleeding or systemic inflammatory response during surgery, volume was administered according to fluid requirements to maintain hemodynamic stability. Blood and blood products were administered as per clinical judgement [20]. Copeptin concentrations were measured preoperatively, within two hours after surgery, and on the first and third postoperative day. All data were recorded and stored in the data management system ‘Clincase’, v2.7.0.12 hosted by IT Systems & Communications, Medical University of Vienna, Vienna, Austria.

### 2.3. Statistical Analysis

We performed an intention-to-treat analysis according to allocated randomisation. Continuous variables were summarised using mean, standard deviation (SD), median, quartiles [25th percentile; 75th percentile] as well as minimum and maximum values. Descriptive statistics are given for randomised groups separately. Categorical variables were summarised using absolute and percent values. Continuous intraoperative values were compared between groups using Mann–Whitney U tests. To investigate a difference in the time course of Copeptin concentrations between the two study groups, first a linear regression model for Copeptin accounting for time, study group and the interaction between time and group as fixed factors as well as accounting for subject ID as random factor was performed. Univariable linear regression models (with random factor subject) were performed for the possible influence factors of time (without interaction term time), type of surgery (open or laparoscopic), age, BMI, sex, ASA physical status, history of coronary artery disease, peripheral artery disease, stroke, heart failure, diabetes, and hypertension. All factors significant (with a *p* < 0.05) in the simple models were then included in a multivariable regression model (with random factor patient). All analyses were performed using R version 3.3.3 and SAS version 9.4 (SAS Institute, Cary, NC, USA).

### 2.4. Post-Hoc Analysis

We compared the perioperative time-course of Copeptin concentrations between patients who developed MINS and patients who did not develop MINS. We measured Troponin T concentrations in all patients preoperatively, within 2 h after surgery, on the first and third postoperative days. MINS was defined as a postoperative Troponin T concentration of 20–65 ng·L^−1^ with an absolute change of at least 5 ng·L^−1^ or a postoperative Troponin T concentration > 65 ng·L^−1^. Patients in whom Troponin T concentration was adjudicated for nonischemic etiology (e.g., sepsis, pulmonary embolism) were not considered as having MINS [24]. We performed a Mann–Whitney U test to compare Copeptin values at each time point. We further performed a receiver-operating characteristics (ROC) curve to evaluate the predictive value of Copeptin and Troponin T concentrations at baseline and within two hours after surgery on the occurrence of postoperative MINS. Furthermore, we performed a ROC curve to investigate the predictive value of Copeptin concentrations in the perioperative period on the occurrence of a composite of postoperative cardiovascular complications, including cardiac failure, myocardial infarction, new onset of cardiac arrhythmias and death.

### 2.5. Sample Size Considerations

Out of the 260 patients planned for the primary aim of the main study, we included 173 patients in our secondary analysis.

We re-estimated the sample size for this secondary analysis based on previous data on Copeptin to get an evaluation of the available sample size. Previous data showed that postoperative Copeptin concentrations in patients undergoing vascular surgery and developing myocardial injury increased up to 100 ± 80 pmol·L^−1^ compared with Copeptin concentrations of 65 ± 80 pmol·L^−1^ in patients without myocardial injury [25]. Therefore, we assumed a difference of 35% in postoperative Copeptin concentrations as clinically meaningful. Using a two-sided t-test, we calculated that at least 82 patients per group are needed to detect a significant difference between both study groups at a significance level of 0.05 with 80% power. Thus, the given sample size of 173 (85 vs. 88) may be adequate to detect the assumed clinically relevant effect.

## 3. Results

A total of 173 consecutive patients, who were enrolled in the main trial from August 2018 to May 2019, were included in this secondary analysis. Eighty-five patients received 80% inspired oxygen and eighty-eight patients received 30% inspired oxygen throughout surgery and for the first two postoperative hours (Figure 1).

Baseline characteristics, including age, weight, ASA physical status, cardiovascular comorbidities, long-term medications and baseline laboratory parameters, were balanced between the two study groups (Table 1). The duration of anesthesia and surgery, anaesthetics, fluid, and vasopressors administered, hemodynamic parameters, and arterial blood gas analyses were balanced between both study groups (Table 2).

### 3.1. Primary Outcome

We did not observe a significant difference in Copeptin concentrations in the overall perioperative time course (*p* = 0.446) between both study groups (Figure 2). Furthermore, at none of the time points was a significant difference in Copeptin concentrations between the 80% oxygen and 30% oxygen group found (2 h postoperative: *p* = 0.090; postoperative day 1: *p* = 0.936; postoperative day 3: *p* = 0.935) (Table 3).

Copeptin concentrations increased significantly within two hours after surgery, compared with baseline values in the overall study population (estimated effect pre vs. post: −241.7 pmol·L^−1^; 95% CI −264.4 to −219.1; *p* < 0.001) as well as in each study group (each *p* < 0.001). Similarly, Copeptin concentrations on the first postoperative day were elevated significantly from baseline values in the overall study population (estimated effect pre vs. post: −35.2 pmol·L^−1^; 95% CI −58.2 to −12.2; *p* = 0.003) as well as in each study group (80% oxygen group: *p* = 0.032; 30% oxygen group: *p* = 0.037). There was no significant difference found in Copeptin concentrations on the third postoperative day compared with baseline values in the overall study population (*p* = 0.505), or the 80% oxygen group (*p* = 0.610) or the 30% oxygen group (*p* = 0.666) separately.

Baseline patient characteristics, including age, sex, BMI, ASA physical status, history of coronary artery disease, history of peripheral artery disease, history of stroke, heart failure, diabetes or hypertension, did not significantly affect perioperative Copeptin concentrations in the univariable regression model (all *p* > 0.05). On an average over all time points, significantly larger Copeptin concentrations were found for open as compared to laparoscopic surgery (*p* = 0.014). A larger increase in Copeptin concentrations from baseline to two hours after surgery was found for open surgeries as compared to laparoscopic surgeries (*p* = 0.001).

In the multivariable regression model, Copeptin values within two hours after surgery were significantly higher in patients receiving open as compared to laparoscopic surgery (*p* < 0.001).

No significant difference was observed in postoperative Copeptin concentrations from baseline to the first or third postoperative day between open or laparoscopic surgeries (Appendix A, Table A1).

### 3.2. Post-Hoc Analysis

We observed significantly higher Copeptin concentrations in patients with MINS as compared to patients without MINS before surgery (14.1 [IQR 8.1 to 22.4] versus 7.7 [IQR 4.5 to 14.2]; *p* = 0.002), on the first postoperative day (49.0 [IQR 29.7 to 116.0] versus 26.7 [IQR 11.9; 53.6]; *p* = 0.002) and on the third postoperative day (24.3 [IQR 16.1 to 46.3] versus 12.5 [IQR 7.2 to 21.5]; *p* = 0.002). Copeptin concentrations within two hours after surgery were similar between patients with MINS (190.3 [IQR 118.2 to 376.9]) and patients without MINS (196.8 [IQR 109.0 to 362.9]) (*p* = 0.840) (Appendix A, Figure A1). Figure A2 in Appendix A shows ROC curves for preoperative Copeptin concentrations and MINS (Area under the curve (AUC) = 0.686; 95% CI 0.586 to 0.926) as well as for preoperative Troponin T concentrations and MINS (AUC = 0.908; 95% CI 0.849 to 0.967) (Appendix A). The area under the ROC curve for Copeptin concentrations within two hours after surgery and MINS was 0.514 (95% CI 0.400 to 0.628) and for Troponin T within two hours after surgery and MINS was 0.480 (95% CI 0.368 to 0.592) (Appendix A, Figure A2).

Overall, 10 patients in this secondary analysis developed a postoperative cardiovascular complication within 30 days after surgery. Copeptin concentrations at baseline (AUC = 0.666; 95% CI 0.445 to 0.886) or within 2 h after surgery (AUC = 0.611; 95% CI 0.432 to 0.789) did not show a predictive value for the occurrence of cardiovascular complications within 30 days after surgery. The area under the ROC curve for Copeptin concentrations on the first postoperative day and cardiovascular complications was 0.819 (95% CI 0.688 to 0.950) and for Copeptin concentrations on the third postoperative day and cardiovascular complications was 0.866 (95% CI 0.743 to 0.988) (Appendix A, Figure A3).

## 4. Discussion

The administration of perioperative supplemental oxygen did not significantly attenuate the release of postoperative Copeptin concentrations in patients at risk for cardiovascular complications undergoing moderate- to high-risk major abdominal surgery. However, we observed a significant increase in postoperative Copeptin concentrations compared with preoperative baseline values in both study groups as well as in the overall study population.

In contrast to the non-surgical setting, we did not observe significant stress reduction in patients who received perioperative supplemental oxygen [27]. One explanation could be that in the study performed in the non-surgical setting, supplemental oxygen was administered for four weeks during the night [27]. Furthermore, only patients with stable heart failure and documented Cheyne–Stokes respiration, who have a high risk for nocturnal desaturation, were included [27]. In this context, the authors suggested that the administration of supplemental oxygen leads to a reduction in episodes of desaturation, which finally leads to reduction in stress [27]. Thus, it might be possible that the duration of oxygen administration in our study was too short to show the same effects. Furthermore, patients undergoing surgery are closely monitored, which makes episodes of desaturation very unlikely. Therefore, our patients might have not been exposed to stress caused by hypoxic events.

An in vitro study has shown that hyperoxia leads to a significant increase in cytotoxicity in adult cardiac myocytes [28]. A retrospective analysis of the PROXI trial has shown that supplemental oxygen increases the risk of myocardial complications after noncardiac surgery [29]. However, a further retrospective sub-analysis of a more recent prospective trial, which investigated the effect of 80% versus 30% oxygen on wound-related complications, did not observe a negative effect of intraoperative supplemental oxygen on cardiovascular complications [30]. More importantly, the most recent trial also showed no negative effects of supplemental oxygen on the incidence of MINS in patients with cardiovascular risk factors undergoing major noncardiac surgery [31]. These findings are consistent with the results of our main trial [21]. Similar to postoperative Troponin T concentrations, the administration of supplemental oxygen also did not result in a significant difference in postoperative Copeptin concentrations.

It has been shown recently that preoperative Copeptin values > 14 ng/L have a high predictive value for the development of myocardial injury after surgery [11]. However, the trend of Copeptin concentrations in the postoperative period was only investigated in a relatively small study on 30 patients undergoing major vascular surgery [25]. In our post-hoc analysis, we observed significantly increased Copeptin concentrations in patients with MINS as compared to patients with no MINS on the first and third postoperative days. Copeptin concentrations within 2 h after surgery did not differ significantly between those groups. Interestingly, we found that Copeptin concentrations before surgery and two hours after surgery were not superior to Troponin T at these time points for predicting MINS. Therefore, it seems likely that Copeptin concentrations in the preoperative and immediate postoperative period do not surpass Troponin T concentrations in the early stratification of patients at risk of developing MINS.

Several studies have shown that noncardiac surgery is associated with a significant postoperative increase in cardiac and stress markers [21,32]. The time after surgery remains a very decisive period associated with cardiovascular complication [2,33]. Troponin T and NT-proBNP concentrations in the first postoperative days are strong predictors for myocardial injury and myocardial strain [2,34]. In contrast to NT-proBNP and Troponin T, which increase approximately 48 h after major abdominal surgery [21,35], we observed that Copeptin concentrations peak within two hours after surgery. Nevertheless, only Copeptin concentrations on the first and third postoperative days were predictive for the development of MINS. Copeptin concentrations have been shown to be significantly elevated in patients experiencing cardiovascular morbidities, including myocardial infarction, stroke and heart failure. Postoperative atherosclerotic complications are the leading cause of postoperative deaths in patients undergoing major noncardiac surgery [24]. Nevertheless, while several risk factors for the development of cardiovascular complications have been established, a clear pathophysiologic explanation has not been determined yet [36]. In our secondary analysis we found a significant increase in postoperative Copeptin concentrations, which highlights the fact that noncardiac surgery is associated with a significant postoperative stress response. Furthermore, we observed a predictive value of preoperative Copeptin concentrations for the development of MINS as well as a predictive value of Copeptin concentrations on the first and third postoperative days for the development of cardiovascular complications. Based on our results, further studies should investigate the impact of perioperative stress on the occurrence of postoperative cardiovascular complications in patients undergoing major noncardiac surgery.

We observed significantly higher postoperative Copeptin concentrations in the overall study population compared with baseline values. Surgery is associated with significantly higher cortisol concentrations, inflammatory response and oxidative stress [37,38,39]. Similar postoperative responses in oxidative stress were also observed in another secondary analysis of our main trial [40]. In detail, we have shown that oxidative stress, assessed via oxidation–reduction potential, which is a reliable marker for oxidative stress [41], significantly increased in the overall study population [40]. Furthermore, there was a simultaneous decrease in the oxidation–reduction capacity [40].

Our study has some limitations. This is a secondary analysis of our main trial [21]. The primary study was powered to detect the effect of supplemental oxygen on postoperative maximum NT-proBNP concentrations [21]. Nevertheless, the given sample size may be adequate to detect clinically relevant effects of supplemental oxygen on Copeptin concentrations. We did not measure further biomarkers to assess perioperative stress such as catecholamines or cortisol concentrations in our study population. The additional assessment of other biomarkers might have provided more substantial information on the effect of supplemental oxygen on perioperative stress response.

## 5. Conclusions

In summary, we showed that the administration of supplemental oxygen has no significant effect on postoperative Copeptin concentrations, which has been used as a surrogate parameter for surgical stress response and myocardial injury. However, we found that Copeptin increased earlier as compared to other biomarkers. Based on our results and previous literature, it is becoming more evident that surgical trauma is a very stressful event, which was reflected by a significant increase in postoperative Copeptin concentrations. In this context, supplemental oxygen might play a negligible role in the postoperative stress response, which could be predominantly caused by surgery.

## Figures and Tables

**Figure 1 jcm-11-02085-f001:**
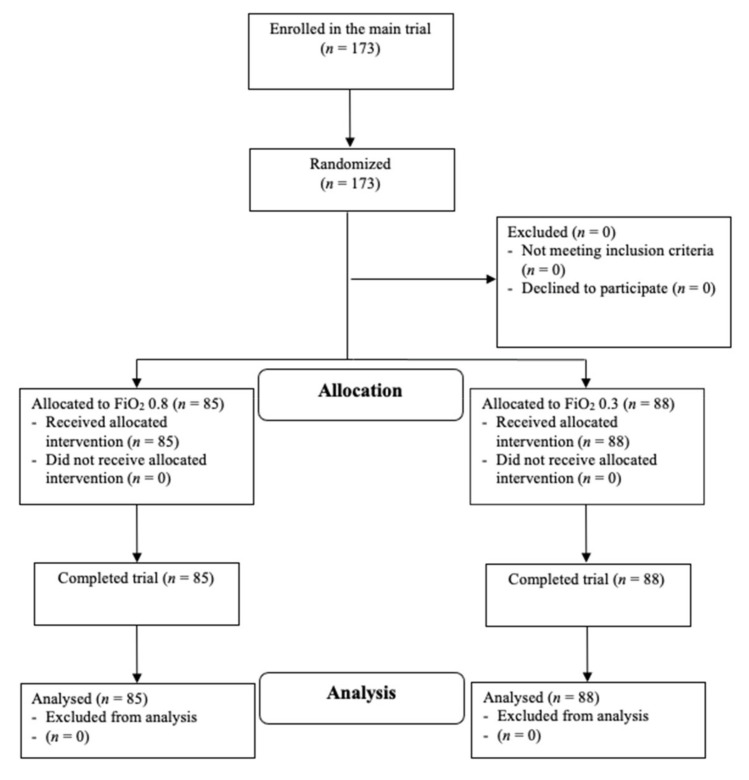
Patient Flow Diagram; Design and Form in Accordance with the 2010 CONSORT Guidelines [26].

**Figure 2 jcm-11-02085-f002:**
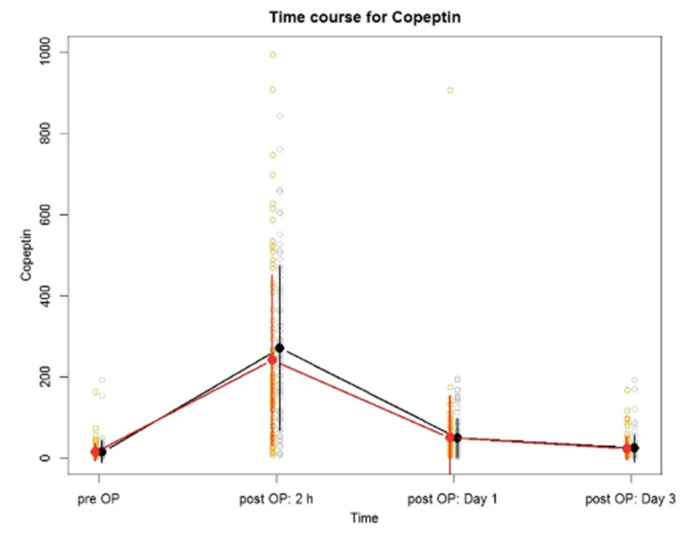
Plot showing the perioperative trend of Copeptin concentrations between patients who received 0.8 FiO_2_ (red) and patients who received 0.3 FiO_2_ (black). Dots represent mean values, vertical lines represent standard deviations of each group. The blank dots give the values of the observed individuals separately for the two groups.

**Table 1 jcm-11-02085-t001:** Patient Characteristics.

	80% Oxygen (*n* = 85)	30% Oxygen (*n* = 88)
Age, years	73 (70; 78)	74 (70; 79)
Height, cm	172 (165; 176)	172 (167; 178)
Weight, kg	80 (67; 93)	75 (67; 90)
BMI, kg·m^−2^	26.6 (23.8; 30.7)	24.9 (23.2; 27.7)
Sex, *n* (%)		
Women	31 (36.5)	28 (31.8)
Men	54 (63.5)	60 (68.2)
ASA physical status, *n* (%)		
II	16 (18.8)	30 (34.1)
III	67 (78.8)	58 (65.9)
IV	2 (2.4)	0 (0)
Comorbidities, *n* (%)		
Hypertension	79 (92.9)	82 (93.2)
Coronary artery disease	24 (28.2)	23 (26.1)
Peripheral artery disease	13 (15.3)	15 (17.0)
Stroke	7 (8.2)	5 (5.7)
Congestive heart failure	5 (5.9)	6 (6.8)
Transient ischemic attack	2 (2.4)	2 (2.3)
Diabetes	26 (30.6)	19 (21.6)
Insulin use	7 (8.2)	2 (2.3)
Long-term medication, *n* (%)		
Beta blockers	44 (51.8)	47 (53.4)
ACI/ARB	45 (52.9)	50 (56.8)
Diuretics	31 (36.5)	26 (29.5)
Statins	33 (38.8)	38 (43.2)
Acetylsalicylic acid	24 (28.2)	30 (34.1)
Oral anticoagulant	31 (36.5)	21 (23.9)
Alpha 2 agonist	3 (3.5)	3 (3.4)
Type of Surgery, (%)		
Hepatobiliary	6 (7.1)	6 (6.8)
Colorectal	18 (21.2)	18 (20.5)
Pancreatic	11 (12.9)	14 (15.9)
Urological	37 (42.1)	34 (40.0)
Gynaecological	6 (7.1)	3 (3.4)
Other	10 (11.8)	10 (11.6)
Open vs. Laparoscopic Surgery, (%)		
Open	51 (60.0)	53 (60.2)
Laparoscopic	30 (35.3)	30 (34.1)
Both ^1^	4 (4.7)	5 (5.7)
Laboratory parameters		
CRP, mg·dL^−1^	0.33 (0.10; 0.82)	0.27 (0.10; 0.91)
Creatinine, mg·dL^−1^	0.9 (0.7; 1.1)	0.9 (0.8; 1.1)
Hemoglobin, g·dL^−1^	12.2 (10.7; 13.2)	12.6 (10.8; 13.9)
Leukocytes, G·L^−1^	5.96 (5.03; 7.72)	5.73 (4.85; 7.76)
NT-proBNP, pg·ml^−1^	205 (88; 486)	218 (102; 796)
Troponin T, ng·L^−1^	13 (8; 19)	13 (9; 21)

Summary characteristics are presented as counts, percentages of patients, and median [25th quartile; 75th quartile]. BMI, body mass index; ASA, American Society of Anaesthesiologists physical status; ACI, angiotensin-converting enzyme inhibitor; ARB, angiotensin receptor blocker; CRP, C-reactive protein; NT-proBNP, N-terminal brain natriuretic peptide. ^1^ Defined as conversion from laparoscopic to open procedure.

**Table 2 jcm-11-02085-t002:** Perioperative variables.

	80% Oxygen (*n* = 85)	30% Oxygen (*n* = 88)	*p*-Value
Intraoperative			
Duration of anesthesia, min	272 (186; 355)	259 (205; 352)	0.622
Duration of surgery, min	221 (141; 307)	200 (142; 292)	0.711
Fluid management			
Crystalloid, mL	2160 (1508; 3386)	2578 (1683; 3339)	0.304
Blood loss, mL	300 (0; 600)	275 (0; 725)	0.610
Urine output, mL	300 (150; 475)	300 (200; 500)	0.417
Anesthesia management			
Fentanyl, mcg	1013 (800; 1463)	1100 (838; 1513)	0.459
Propofol, mg	120 (93; 150)	125 (50; 200)	0.536
Phenylephrine, mg	0.28 (0.09; 0.46)	0.21 (0.08; 0.42)	0.717
Noradrenaline, mg	0.25 (0.00; 0.60)	0.20 (0.00; 0.08)	0.491
etSevo, %	1.3 (1.0; 1.3)	1.2 (1.0; 1.3)	0.556
FiO_2_, %	80 (80; 80)	31 (30; 32)	
etCO_2_, mmHg	34 (32; 36)	34 (31; 35)	0.531
Core temp, °C	36.5 (36.1; 36.8)	36.5 (36.2; 36.9)	0.210
Hemodynamic Parameters			
HR, beats·min^−1^	70 (58; 86)	65 (56; 73)	0.845
MAP, mmHg	80 (76; 84)	81 (76; 88)	0.549
SV, mL	71 (63; 84)	66 (57; 83)	0.821
CO, L·min^−1^	4.1 (3.7; 5.6)	4.6 (3.7; 5.3)	0.615
CVP, mmHg	12 (10; 15)	10 (9; 12)	0.086
Arterial Blood Gas Analysis			
pO_2_, mmHg	314 (270; 361)	131 (109; 158)	<0.001
pCO_2_, mmHg	42 (40; 44)	41 (39; 43)	0.015
pH	7.38 (7.35; 7.41)	7.39 (7.35; 7.42)	0.169
BE	−0.6 (−1.9; 0.9)	−0.3 (−1.9; 0.9)	0.765
Hemoglobin, g·dL^−1^	11.7 (9.9; 12.8)	11.7 (10.2; 12.9)	0.745
Lactate, mmol·L^−1^	0.9 (0.7; 1.2)	0.9 (0.7; 1.1)	0.745
Glucose, mmol·L^−1^	7.3 (6.4; 8.9)	7.0 (6.2; 8.1)	0.071
2 h postoperative			
Hemodynamic Parameters			
HR, beats·min^−1^	75 (61; 91)	69 (63; 77)	0.450
MAP, mmHg	82 (76; 100)	81 (77; 100)	0.431
72 h postoperative			
Fluid, mL ^a^	9852 (6845; 11,989)	9506 (7200; 12,137)	0.900
Piritramide, mg ^b^	8.0 (3.0; 20.3)	10.0 (3.0; 21.0)	0.903

Summary characteristics of perioperative variables are presented as medians [25th quartile; 75th quartile]. All *p*-values are for two-tailed Mann–Whitney U tests. etSevo, end-tidal Sevoflurane concentration; FiO_2_, Fraction of inspired oxygen; etCO_2_, end-tidal carbon dioxide concentration; HR, heart rate; MAP, mean arterial pressure; SV, stroke volume; CO, cardiac output; CVP, central venous pressure; pO_2_, oxygen partial pressure; pCO_2_, carbon dioxide partial pressure; BE, base excess. ^a^ Overall amount of fluid administered during the first 72 h after surgery. ^b^ Overall amount of piritramide administered during the first 72 h after surgery.

**Table 3 jcm-11-02085-t003:** Univariable regression model Copeptin.

Variable	Comparison	Effect	Lower CI	Upper CI	*p*-Value
Time	pre vs. 2 h post	−241.740	−264.440	−219.050	<0.001
	pre vs. POD 1	−35.206	−58.245	−12.168	0.003
	pre vs. POD 3	−7.976	−31.470	15.519	0.505
Time × Group	Group 30% vs. 80% pre	−0.087	−35.119	34.944	0.996
	Group 30% vs. 80% 2 h post	29.838	−4.614	64.291	0.090
	Group 30% vs. 80% POD 1	−1.453	−36.794	33.888	0.936
	Group 30% vs. 80% POD 3	−1.514	−38.053	35.025	0.935
	Group 30% pre vs. 2 h post	−256.380	−288.210	−224.550	<0.001
	Group 30% pre vs. POD 1	−34.513	−66.983	−2.042	0.037
	Group 30% pre vs. POD 3	−7.259	−40.295	25.778	0.666
	Group 80% pre vs. 2 h post	−226.450	−258.820	−194.080	<0.001
	Group 80% pre vs. POD 1	−35.878	−68.582	−3.175	0.032
	Group 80% pre vs. POD 3	−8.685	−42.115	24.744	0.610
Type of surgery	Laparoscopic vs. Open	−31.164	−55.866	−6.463	0.014
Time × Type of surgery	Overall Interaction				<0.001
Age		0.559	−0.937	2.055	0.464
BMI		0.194	−2.136	2.523	0.871
Sex	Female vs. Male	15.426	−9.309	40.161	0.221
ASA	III, IV vs. I, II	−3.227	−29.836	23.383	0.812
Coronary Artery Disease	Yes vs. No	5.535	−20.697	31.767	0.679
Peripheral Artery Disease	Yes vs. No	19.548	−12.763	51.859	0.235
Stroke	Yes vs. No	1.759	−44.601	48.118	0.941
Heart Failure	Yes vs. No	22.391	−24.925	69.707	0.353
Diabetes	Yes vs. No	−1.582	−28.143	24.980	0.907
Hypertension	Yes vs. No	−4.915	−51.273	41.443	0.835

The estimated effect sizes, confidence intervals (CI) and *p*-values were calculated using univariable regression models. pre, preoperative; 2 h post, within two hours after surgery; POD, postoperative day; BMI, body mass index; ASA, American Society of Anesthesiologists.

## Data Availability

The data presented in this secondary analysis are available on request from the corresponding author.

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
