# Peer review of "Perioperative Supplemental Oxygen and Postoperative Copeptin Concentrations in Cardiac-Risk Patients Undergoing Major Abdominal Surgery—A Secondary Analysis of a Randomized Clinical Trial"

_jcm, 2022, doi:10.3390/jcm11082085_

Round 1
Reviewer 1 Report
I consider that the idea of this study, to evaluate the effect of 80% versus 30% perioperative oxygen administration on postoperative maximum NT-proBNP concentrations and myocardial injury after noncardiac surgery is very interesting and with important clinical practice, considering that Copeptin concentrations accurately reflect myocardial strain and injury as well as endogenous stress. The article is well-structured, the results are presented in an appropriate manner, but I suggest emphasizing much better the results by referring to other studies, in order to highlight the novelty of the study. Also, I suggest including the details of the broader impacts on the study made. I would like to ask the authors if they took into account the other comorbidities of the patients, or if they excluded patients with some specific pathologies?
Author Response
Reviewer 1:
I consider that the idea of this study, to evaluate the effect of 80% versus 30% perioperative oxygen administration on postoperative maximum NT-proBNP concentrations and myocardial injury after noncardiac surgery is very interesting and with important clinical practice, considering that Copeptin concentrations accurately reflect myocardial strain and injury as well as endogenous stress. The article is well-structured, the results are presented in an appropriate manner, but I suggest emphasizing much better the results by referring to other studies, in order to highlight the novelty of the study. Also, I suggest including the details of the broader impacts on the study made.
We thank the reviewer for the comments. We completely agree with the reviewer and added the following paragraph in the discussion section:
“Copeptin concentrations have been shown to be significantly elevated in patients experiencing cardiovascular morbidities including myocardial infarction, stroke and heart failure. Postoperative atherosclerotic complications are the leading cause of postoperative deaths in patients undergoing major noncardiac surgery [41]. Nevertheless, while several risk factors for the development of cardiovascular complications have been established, a clear pathophysiologic explanation has not been determined yet [42]. In our secondary analysis we found a significant increase in postoperative Copeptin concentrations, which highlights the fact that noncardiac surgery is associated with a significant postoperative stress response. Furthermore, we observed a predictive value of preoperative Copeptin concentrations for the development of MINS as well as a predictive value of Copeptin concentrations on the first and third postoperative days for the development of cardiovascular complications. Based on our results, further studies should investigate the impact of perioperative stress on the occurrence of postoperative cardiovascular complications in patients undergoing major noncardiac surgery.”
I would like to ask the authors if they took into account the other comorbidities of the patients, or if they excluded patients with some specific pathologies?
We included patients with pre-existing comorbidities (history of coronary artery disease, history of peripheral artery disease, stroke, heart failure, diabetes, hypertension). In a univariate regression model, none of the aforementioned cardiovascular comorbidities significantly affected postoperative Copeptin concentrations. Patients with the following pathologies were excluded from this study: sepsis, preoperative inotropic therapy, oxygen dependent patients, severe heart failure (defined as recorded ejection fraction <30%).

Reviewer 2 Report
Since copeptin and troponin are used for earlier diagnosis of myocardial infarction, the question arises whether there were changes in the ECG before and after surgery. Was this analyzed?
Was there a difference in the concentrations in patients who had coronary artery disease or bypass surgery and those who did not?
How many of the patients had pneumonia? Were there significant differences in copeptin levels here?
The authors give only a sketchy information about perioperative hypovolemia, hypotension and tachycardia, which according to the authors (line 44-46) influence the change of the studied parameter. Is it possible to specify this information?
What consequences would changes in copeptin levels have for OP management?
The authors state here to reproduce a subgroup analysis of a previously published paper in which the correlation to NT-proBNP was studied. Since no significant difference was apparently found in the main paper either, the question arises why the change in copeptin levels was not also picked up in the previously published paper. In other words, should we expect another subgroup analysis comparing another parameter between the groups and showing no difference?
Did the authors investigate whether combining troponin and copeptin levels would result in a significant finding?
How did the authors quantify myocardial injury after noncardiac surgery? For example, were MRI examinations performed here? Or echocardiographic controls?
The authors should better elaborate on the role of copeptin, which is a prohormone localized to the hypothalamus and influenced more by volume and blood pressure fluctuations than by coronary underperfusion (where troponin increases).
Were measurement methods also used to measure the sedation depth or the operative stress (this is not evident from the tables)?
Essential elements that are also missing in the tables is the information about lung disease such as COPD and asthma. Also, the NYHA classification before and after surgery and a graduation of the angina pectoris symptomatology of the patients should be reflected. The indication with heart failurel yes yes or no is not sufficient. Data on left ventricular (EF) systolic function are missing.
How many of the patients studied developed a cardiovascular complication? Were there correlations here with the parameters studied?
Would it make sense to intensify the intervals of the determination of the copeptin values perioperatively?
Unfortunately, it is not clear from the paper which endpoints were chosen. The sole determination of a laboratory parameter, which is investigated in correlation to myocardial events in patients with increased risk, it is inevitable to reflect also the actually occurred cardiovascular events or to correlate the investigated parameter with them.
It is also not clearly worked out why a preoperative administration of oxygen should influence a parameter which is rather influenced by the displacement of body fluids.
Author Response
Reviewer 2:
Since copeptin and troponin are used for earlier diagnosis of myocardial infarction, the question arises whether there were changes in the ECG before and after surgery. Was this analyzed?
As defined by the study protocol (Reiterer C, Trials 2020) myocardial infarction was defined according to the 4th universal definition of myocardial infarction (Thygesen K, J Am Coll Cardiol, 2018). Myocardial infarction is defined as a rise of Troponin T above the 99th percentile upper range limit and at least one of the following criteria: symptoms of myocardial ischemia, new ischemic ECG changes, development of pathological Q waves, imaging evidence of new loss of viable myocardium or identification of a coronary thrombus by angiography (Thygesen K, J Am Coll Cardiol, 2018). One patient in the 80% oxygen group and no patient in the 30% oxygen group developed myocardial infarction. All additional ECG were performed by the attending physicians as per clinical judgement during hospitalization.
Was there a difference in the concentrations in patients who had coronary artery disease or bypass surgery and those who did not?
We performed a univariable regression model to evaluate a possible effect of pre-existing coronary artery disease on postoperative Copeptin concentrations. We found no significant effect in patients with history of coronary artery disease on postoperative Copeptin concentrations (estimated effect: 5.535; CI -20.697 to 31.767; p = 0.679) (Table 3).
How many of the patients had pneumonia? Were there significant differences in copeptin levels here?
Four patients included in this secondary analysis developed postoperative pneumonia (three patients in the 30% oxygen group; one patient in the 80% oxygen group). To evaluate a possible difference in perioperative Copeptin concentrations between patients, who developed pneumonia and patients without pneumonia, we performed a post-hoc Mann-Whitney-U. We found no significant differences in patients with or without pneumonia at baseline (p = 0.160), within 2 hours after surgery (p = 0.707), on the first postoperative day (p = 0.974) or on the third postoperative day (p = 0.223). Since only four patients developed pneumonia, we did not include this analysis in our manuscript. If the reviewer deems it necessary, we will include this analysis in our manuscript.
The authors give only a sketchy information about perioperative hypovolemia, hypotension and tachycardia, which according to the authors (line 44-46) influence the change of the studied parameter. Is it possible to specify this information?
We thank the reviewer for pointing this out. We added the following paragraph in the methods section of the manuscript:
“Intraoperative fluid management in all patients was performed esophageal-Doppler-guided according to a previously published algorithm [37, 38]. As per study protocol, all patients received a 2ml.kg-1.BW-1 baseline infusion of balanced crystalloids. A bolus of 250ml balanced crystalloids was administered when stroke volume decreased by ≥20% from baseline. In the case of acute bleeding or systemic inflammatory response during surgery, volume was administered according to fluid requirements to maintain hemodynamic stability. Blood and blood products were administered per clinical judgement.”
What consequences would changes in copeptin levels have for OP management?
In this secondary analysis we observed that besides the known predictive value of preoperative Troponin T concentration, preoperative Copeptin also had a significant predictive value for the incidence of MINS. Patients developing MINS are at increased risk for cardiovascular complications within 30 days after surgery and have a significantly higher mortality (Botto F, Anesthesiology 2014). It has been shown previously that even short periods of intraoperative hypotension are significantly associated with an increased incidence of MINS (Abbott, Anesth Analg 2018). Therefore, in patients with elevated baseline concentrations of Copeptin, intraoperative blood pressure management should be performed with particular attention at avoidance of hypotension.
The authors state here to reproduce a subgroup analysis of a previously published paper in which the correlation to NT-proBNP was studied. Since no significant difference was apparently found in the main paper either, the question arises why the change in copeptin levels was not also picked up in the previously published paper. In other words, should we expect another subgroup analysis comparing another parameter between the groups and showing no difference?
We totally agree with the reviewer. However, Copeptin addresses the effect of physiological stress as compared to NT-proBNP, which is a strong biomarker for cardiac function. To fully address the effect of Copeptin we planned to publish the results in a separate paper.
Patients undergoing major noncardiac surgery often experience cardiovascular complications because of increased myocardial strain, which is mostly caused by a mismatch in oxygen supply and demand (Priebe HJ, Br J Anaesth, 2004; Devereaux PJ, CMAJ, 2005). Several retrospective studies have investigated possible effects of supplemental oxygen on the incidence of cardiovascular complications with inconsistent results (Fonnes S, Int J Cardiol, 2016; Ruetzler K, Anesth Analg, 2019). It was shown previously that the administration of supplemental oxygen leads to a decrease in heart rate and subsequently reduces myocardial oxygen consumption (Smit B, Crit Care, 2018). To investigate the effect of supplemental oxygen on myocardial strain – assessed via consecutive NT-proBNP concentrations – we performed the main trial in which we found no significant difference in postoperative NT-proBNP concentrations between both study groups.
We assessed perioperative Copeptin concentrations not to assess myocardial strain but rather the postoperative stress response in patients at-risk for cardiovascular complications undergoing major noncardiac surgery. It was shown previously that the administration of supplemental oxygen led to a reduction in plasma catecholamine concentrations (Staniforth A, Eur Heart J, 1998). Therefore, we hypothesized that perioperative administration of supplemental oxygen might lead to a reduction in the postoperative stress response. Because of the different pathophysiologic effects of supplemental oxygen investigated, we decided not to include the analysis of perioperative Copeptin concentrations between both study groups in the main trial.
Did the authors investigate whether combining troponin and copeptin levels would result in a significant finding?
In our post-hoc analysis we evaluated perioperative Copeptin concentrations between patients with MINS and patients without MINS. We found significantly higher Copeptin concentrations in patients with MINS as compared to patients without MINS at baseline, on the first and on the third postoperative day. Furthermore, we found a significant predictive value of preoperative Copeptin concentrations for the development of MINS. However, preoperative Troponin T concentrations had an even higher predictive value for MINS. We added a post-hoc analysis, in which we evaluated the predictive value of Copeptin concentrations in the perioperative time course for the development of postoperative cardiovascular complications. We found that Copeptin concentrations on the first and third postoperative days were predictive for postoperative cardiovascular complications. However, our study was not powered to accurately evaluate the association between perioperative Copeptin concentrations and postoperative cardiovascular complications. Currently, we are conducting a multi-center prospective observational study to extend our knowledge on the perioperative course of cardiovascular and inflammatory biomarkers and their association with the incidence of postoperative cardiovascular complications in patients undergoing noncardiac surgery (ClinicalTrials.gov NCT04753307).
How did the authors quantify myocardial injury after noncardiac surgery? For example, were MRI examinations performed here? Or echocardiographic controls?
We used the current definition for diagnosing MINS, which is by consecutive Troponin T measurements. We did not perform MRI or echocardiographic examinations. We added the current MINS definition in our manuscript at the method section.
“We measured Troponin T concentrations in all patients preoperatively, within 2 hours after surgery, on the first and third postoperative day. MINS was defined as a postoperative Troponin T concentration of 20 – 65 ng.L-1 with an absolute change of at least 5 ng.L-1 or a postoperative Troponin T concentration > 65 ng.L-1. Patients, in whom Troponin T concentration was adjudicated for nonischemic etiology (e.g. sepsis, pulmonary embolism) were not considered as having MINS [36].”
The authors should better elaborate on the role of copeptin, which is a prohormone localized to the hypothalamus and influenced more by volume and blood pressure fluctuations than by coronary underperfusion (where troponin increases).
We agree with the reviewer. We changed the introduction of the manuscript:
“Copeptin is a relatively novel biomarker and reflects plasma concentrations of arginine-vasopressin (AVP) [11]. AVP is an antidiuretic hormone released from the hypothalamus in response to changes in plasma osmolality and blood pressure and its main function is homeostasis of fluid balance, vascular tonus and regulation of the endocrine stress response [39]. In detail, an increase in blood osmolality and hypovolemia lead to increased plasma AVP and Copeptin concentrations [40]. In contrast to AVP, plasma concentrations of Copeptin are very stable and simple to measure and therefore used to indirectly assess plasma AVP concentration [11].”
Were measurement methods also used to measure the sedation depth or the operative stress (this is not evident from the tables)?
We maintained anesthesia guided by processed electro-encephalogram (EEG) according to our study protocol. (Reiterer C, Trials 2020). In detail, anesthesia was maintained with Sevoflurane up to a minimal alveolar concentration of 1.5% in an oxygen gas carrier according to guided anesthesia. We presented the expiratory concentrations of sevoflurane in both groups, which were similar between both study groups, in Table 2. Unfortunately, EEG data was not possible to be extracted from our electronic anesthesia protocol, therefore, we are not able to present the data.
Essential elements that are also missing in the tables is the information about lung disease such as COPD and asthma. Also, the NYHA classification before and after surgery and a graduation of the angina pectoris symptomatology of the patients should be reflected. The indication with heart failurel yes yes or no is not sufficient. Data on left ventricular (EF) systolic function are missing.
History of COPD or Asthma was defined as a previous diagnosis of COPD/Asthma by a physician with the appropriate ICD-10 code. Unfortunately, we did not assess NYHA classification before and after surgery since this is not standard of clinical assessment in our center. We only included patients with a history of angina pectoris symptoms but no patients with angina symptoms at time of surgery. Patients with actual symptoms did not undergo surgery until full cardiovascular assessment was performed by a cardiologist. ECG and further diagnostics were performed by the attending physicians per clinical judgement. As defined by the study protocol (Reiterer C, Trials, 2020) postoperative heart failure was defined as cardiac decompensation requiring inotropic or vasopressor therapy. Overall, four patients in this study population developed postoperative cardiac failure requiring inotropic therapy.
How many of the patients studied developed a cardiovascular complication? Were there correlations here with the parameters studied?
We thank the reviewer for the interesting question. To investigate Copeptin concentrations in patients, who developed cardiovascular complications, we performed an additional post-hoc analysis. We added the following sentence in the Methods Section:
“Furthermore, we performed a ROC curve to investigate the predictive value of Copeptin concentrations in the perioperative period on the occurrence of a composite of postoperative cardiovascular complications including cardiac failure, myocardial infarction, new onset of cardiac arrhythmias and death.”
And the following paragraph in the results section:
“Overall, ten patients in this secondary analysis developed a postoperative cardiovascular complication within 30 days after surgery. Copeptin concentrations at baseline (AUC=0.666; 95% CI 0.445 to 0.886) or within 2 hours after surgery (AUC=0.611; 95% CI 0.432 to 0.789) did not show a predictive value for the occurrence of cardiovascular complications within 30 days after surgery. The area under the ROC curve for Copeptin concentrations on the first postoperative day and cardiovascular complications was 0.819 (95% CI 0.688 to 0.950) and for Copeptin concentrations on the third postoperative day and cardiovascular complications was 0.866 (95% CI 0.743 to 0.988) (Appendix A, Figure A3).”
Unfortunately, our study was not powered to accurately reflect the association between perioperative Copeptin concentrations and the occurrence of postoperative cardiovascular complications. It remains unclear, whether elevated Copeptin concentrations are at least partially causing cardiovascular complications or if Copeptin concentrations increase as a by-product of cardiovascular complications, which per se lead to an exacerbated stress response.
Would it make sense to intensify the intervals of the determination of the copeptin values perioperatively?
A previous study closely evaluated the perioperative time course of Copeptin concentrations in patients, who underwent cardiac surgery in short postoperative intervals (Holm J, J. Cardiothorac Vasc Anesth, 2018). Similarly, to our results, they found maximum Copeptin concentrations immediately after surgery and a rapid decrease in Copeptin concentrations in the following postoperative hours and days. In our post-hoc analysis we found a predictive value of Copeptin concentrations on the first and third postoperative days for the occurrence of cardiovascular complications within 30 days after noncardiac surgery. However, our study was not powered to accurately evaluate the association between perioperative Copeptin concentrations and postoperative cardiovascular complications. Therefore, future studies including high number of patients should investigate the effect of the perioperative time course of Copeptin concentrations on the incidence of cardiovascular complications and its use for the perioperative evaluation of patients at risk of developing cardiovascular complications after noncardiac surgery.
Unfortunately, it is not clear from the paper which endpoints were chosen. The sole determination of a laboratory parameter, which is investigated in correlation to myocardial events in patients with increased risk, it is inevitable to reflect also the actually occurred cardiovascular events or to correlate the investigated parameter with them.
We agree with the reviewer. Primary outcome of this secondary analysis was a possible effect of perioperative supplemental oxygen on Copeptin concentrations in the perioperative time course. In our post-hoc analysis we evaluated Copeptin concentrations between patients with MINS and patients without MINS and found significantly higher Copeptin concentrations in patients with MINS. To investigate Copeptin concentrations in patients, who developed cardiovascular complications within 30 days after surgery, we performed an additional post-hoc analysis (see above). Unfortunately, our study was not powered to significantly detect a possible association between of perioperative Copeptin concentrations and postoperative cardiovascular complications. We added a subheading “Primary Outcome” in the results section to make it more clear which analysis was our primary outcome.
It is also not clearly worked out why a preoperative administration of oxygen should influence a parameter which is rather influenced by the displacement of body fluids.
Arginine-vasopressin and Copeptin are significantly elevated in pathophysiologic states of stress due to osmotic, hemodynamic or unspecific stress-related stimuli (Christ-Crain M, Rev Endocr Metab Disord, 2019). Major abdominal surgery is associated with perioperative hemodynamic perturbations, fluid shifts, stress and hypoxic events (Turan A, Anesthesiology, 2019; Shoemaker WC, Crit Care Med, 1988). It was even shown in a previous retrospective cohort study showed that 20% of patients experience intraoperative hypoxemia defined as SpO2 < 90% (Kendale SM, J Clin Anesth, 2016). Furthermore, it was shown previously that hypoxemia in the immediate postoperative period is common and often not recognized by nursing staff in the postoperative care unit (Sun Z, Anesth Analg, 2015). In a previous trial in the non-surgical setting, the administration of supplemental oxygen led to a reduction in plasma catecholamine concentrations (Staniforth A, Eur Heart J, 1998). The author suggested that the administration of oxygen might reduce the incidence of hypoxemia and might therefore lead to a decrease hypoxemic associated stress response. Therefore, we also assumed that perioperative administration of supplemental oxygen might lead to a reduction in episodes of hypoxemia and might therefore lead ot a reduction of postoperative stress as well.

Round 2
Reviewer 2 Report
However, i do not really understand now if there is a difference and effect of oxygenationlavels and Copeptin or not
And again, i think cardiovascular events are not well pointed out. for eg heart failure is not defined as use of inotropes or else after surgery. This can also lead to fluid-loss during surgery ore interactions to anesthetics. However, classification of heart failure is defined by worsening of LVEF or as symptomatic by NYHA classification which should be figured out.
Why do you want to present your findings to ntPro BNP in an further publication? Why don't you combine it with this paper which maybe make this more interesting?
